# The preventative effects of statin on lung cancer development in patients with idiopathic pulmonary fibrosis using the National Health Insurance Service Database in Korea

Yoo Jung Lee[1], Nayoon Kang[2], Junghyun Nam[3], Eung Gu Lee[4], Jiwon Ryoo[4], Soon Seog Kwon[4], Yong Hyun Kim[4], Hye Seon Kang[4]*

1 Department of Internal Medicine, Division of Pulmonary, Allergy and Critical Care Medicine, Incheon St. Mary's Hospital, College of Medicine, The Catholic University of Korea, Incheon, Republic of Korea, 2 Department of Statistics and Data Science, Yonsei University, Seoul, Republic of Korea, 3 Department of Internal Medicine, Division of Respiratory, Allergy and Critical Care Medicine, Seoul St. Mary's Hospital, College of Medicine, The Catholic University of Korea, Seoul, Republic of Korea, 4 Department of Internal Medicine, Division of Respiratory, Allergy and Critical Care Medicine, Bucheon St. Mary's Hospital, College of Medicine, The Catholic University of Korea, Bucheon, Republic of Korea

* beyer_kr@catholic.ac.kr

**Data Availability Statement:** Publicly available datasets were analyzed in this study. This data can be found here: NHIS sharing service websites

## Abstract

Little is known about the effect of statin use in lung cancer development in idiopathic pulmonary fibrosis (IPF). We analyzed the database of the National Health Insurance Service to further investigate the clinical impacts of statin on lung cancer development and overall survival (OS) in IPF patients. The analysis included 9,182 individuals diagnosed with IPF, of which 3,372 (36.7%) were statin users. Compared to statin non-users, the time from diagnosis of IPF to lung cancer development and OS were longer in statin users in IPF patients. In Cox proportional hazard regression models, higher statin compliance, statin use, and being female had an inverse association with lung cancer risk, while older age at diagnosis of IPF and smoking history were associated with higher risk of lung cancer in IPF patients. For OS, statin use, female sex, higher physical activity frequency, and diabetes were associated with longer survival. In contrast, older age at diagnosis of IPF and smoking history were associated with shorter OS in IPF patients. These data from a large population indicate that statin had an independent protective association with lung cancer development and mortality in IPF patients.

## Introduction

Statins are 3-hydroxy-3-methylglutaryl-coenzyme A (HMG-CoA) reductase inhibitors that competitively block the active sites of cholesterol-producing enzymes [1]. The main therapeutic effect is inhibition of cholesterol biosynthesis. However, other diverse actions called

([http://nhiss.nhis.or.kr](http://nhiss.nhis.or.kr), accessed on 15 April 2022).

**Funding:** This work was supported by the Institute of Clinical Medicine Research of Bucheon St. Mary's Hospital, Research Fund, 2020 (BCMC20BD07). here was no additional external funding received for this study. The funders had no role in study design, data collection and analysis, decision to publish, or preparation of the manuscript.

**Competing interests:** The authors have declared that no competing interests exist.

"pleiotropy" were reported, including improvement of cardiovascular function [2], anti-inflammatory effects [3], and anti-fibrotic effects [4,5].

Some studies have indicated that statins exert anticancer effects by inducing apoptosis and inhibiting tumor cell growth and angiogenesis [6–8]. Recent research suggests antitumor effects of statin [9–12] and its synergistic potency in chemo-resistant lung cancer populations [13–16].

Clinically, statin is associated with reduced all-cause mortality in interstitial lung disease and idiopathic pulmonary fibrosis (IPF) [17,18]. Also, statin attenuates decline in lung function in the elderly [19]. Lung cancer is a common complication of IPF [20,21], with an incidence of approximately 22.9 per 10,000 person-years, which is approximately five times that in the general population. Kim et al. published a study showing that IPF patients with lung cancer had poor 5-year survival rates compared to non-IPF patients (14.5% vs. 30.1%; $P<0.001$) [22]. Moreover, there was higher tendency of treatment-related adverse events [23] such as postoperative clinical deterioration, acute exacerbation (AE), and radiation pneumonitis [24–26] among IPF patients with lung cancer. Therefore, the importance of lung cancer prevention in IPF patients is very high.

To date, few studies have examined the role of statins in the risk of lung cancer development among IPF patients in large-scale cohorts. We analyzed the database of the National Health Insurance Service (NHIS) in Republic of Korea to further investigate the clinical impacts of consecutive statin use on lung cancer development and OS in IPF patients.

## Materials and methods

### Study database

This study collected data from NHIS, which is based on a nationwide social security system with more than 50 years of history in Korea. Nearly all Korean citizens (97.2%, ~ 50 million) are enrolled in the NHIS, and data of the study population include demographics, medical treatment, and disease diagnosis according to the International Classification of Diseases, 10th Revision (ICD-10) [27]. The NHIS dataset includes all inpatient and outpatient medical claims and the corresponding codes for diagnoses and treatment procedures [28]. We collected and retrospectively reviewed IPF patients among adults older than 40 years during the study period between 2002 and 2018 with a two-year washout period, therefore the patients diagnosed with IPF from 2002 to 2004 were excluded. The last date we had access to the database was February 10th, 2022.

IPF was defined as a patient recorded with ICD-10 code of J841 or J848, who visited at an outpatient or inpatient clinic at least twice in a year. Code J84 was classified as a rare intractable disease category in Korea. The government supports these patients with medical cost reduction of up to 10% of the total cost; therefore, the ICD for this disease indicates IPF patients retained in the database [29]. Exclusion criteria were as follows: the patients with possible interstitial lung diseases or other systemic involvement with lung disease codes as follows: M05.1, M05.2, M05.3, M05.8, M05.9, M06.0, M06.8, M06.9, M30.1, M31.3, M31.7, M32, M33, M34, M35.0, M35.1, D86, J84.0, J60~J70.9 (S1 Appendix); the patients diagnosed with lung cancer (ICD-10 code of C33 or C34) within a year before the diagnosis of IPF; the patients with pre-existing lung cancer before clinically proven IPF diagnosis during the washout period; and the patients prescribed statins before IPF diagnosis.

The study was conducted according to the guidelines of the Declaration of Helsinki, and approved by the Institutional Review Board of The Catholic University of Korea (IRB no: HC20ZISI0006). Informed consent was waived because data analysis was performed retrospectively using anonymized data derived from the NHIS in Korea.

## Endpoints and study outcomes

The primary endpoint of this study was the duration to develop of lung cancer in IPF patients according to statin use during the study period. This was calculated by subtracting the date of first IPF diagnosis from first lung cancer diagnosis. The secondary endpoint was the OS of IPF patients according to statin use. OS was the duration from the day of first IPF diagnosis to death or the study end date.

## Statin use

We defined a statin user as a patient who had been started to be prescribed statin after the diagnosis of IPF during the study period. The regular consecutive use of statin was defined whether or not he or she prescribed at least four consecutive weeks (28 days). The calculation of statin compliance was conducted using a mathematical approach. Patients were included in the regular statin group if the interval between prescriptions was smaller than the sum of the total prescription days plus 14 days. In cases where the total prescription days were zero or unidentified, they were substituted with the total number of days of statin administration. The study did not include any patients prescribed statin before diagnosis of IPF to exclude conditional bias of statin use. Drug compliance was estimated as the ratio of total dates of prescription to the study period.

## Exposure related factors—Drinking amount at a time

The Korea National Health and Nutrition Examination Survey (KNHANES), a comprehensive nationwide survey periodically conducted by the Korea Centers for Disease Control and Prevention (KCDC) to assess the health status of the South Korean population, data is collected through interviews on health and nutrition, as well as basic health assessments [30]. Within this survey questions asked about the amount of alcohol consumed at each drinking session, for which the possible responses were as follows: (1) 1–2 glasses, (2) 3–4 glasses, (3) 5–6 glasses, (4) 7–9 glasses, and (5) 10 glasses or more. In the survey questions of the KNHANES, 1 glass of alcoholic beverage was explained with the example of 1 glass of Soju, the standard drink in Korea. The Korean alcohol consumption guidelines define 14 g of pure alcohol as 1 standard drink. A 1/4 bottle (approximately 90mL) of 20% Soju may contain approximately 14g of alcohol, equivalent to 1 standard drink [31]. A glass refers to a drink in our study.

## Exposure related factors—Physical activity intensity

Regular physical activity(PA) provides important health benefits for those with chronic health conditions or disabilities, including cancer survivors and people with osteoarthritis, hypertension, type 2 diabetes, multiple sclerosis, stroke, Parkinson's Disease, spinal cord injury, dementia, and other cognitive disorders [32]. Among the variable clinical guidelines of PA intensity, we defined the PA both in absolute and in relative method. The one was by counting the days of activities per week: (1) 1–2 times, (2) 3–4 times, (3) 5–6 times, (4) 7 days per week, the other was by calculating with the unit of metabolic equivalent (MET). A person's MET is three to six times higher when moderately active (3–6 METs) and vigorous active meant more than six times higher when clinically active (>6 METs) [33].

## Statistical analysis

The paired t-test was used for continuous data, and the chi-square test was used for categorical data to generate descriptive tables. Survival was analyzed using a Kaplan–Meier plot and the log-rank test. Kaplan-Meier analysis was used to visualize the duration between lung cancer

diagnosis and death according to statin use. Log rank test was used to validate the significant difference of time-to-event between statin and non-statin user groups. Cox proportional hazard regression analysis was used to adjust for multiple variables affecting the hazard of lung cancer development with a 95% confidence interval (CI). All P-values were two-tailed, with statistical significance set to $P < 0.05$. All statistical analyses were performed using SAS Version 7.1 (SAS Institute, Inc., Cary, NC, USA) and R Version 4.0.3 (R Foundation for Statistical Computing, Vienna, Austria).

## Results

The final analysis included 9,182 individuals diagnosed with IPF, of which 3,372 (36.7%) were statin users. The baseline characteristics of study patients are listed in Table 1. The age at first diagnosis of IPF was younger in the statin user group (67.2 ± 11.2 vs. 64.0 ± 10.2, $P < 0.001$). The proportion of males was higher in the statin non-user group (67.1% vs. 59.4%, $P < 0.001$). Mean body mass index (BMI) (23.0 ± 3.2 vs. 24.0 ± 3.1, $P < 0.001$), total cholesterol (184.2 ± 34.8 vs. 199.7 ± 42.6, $P < 0.0001$), and systolic (125.2 ± 17.1 vs. 126.6 ± 16.8, $P < 0.0001$) and diastolic blood pressure (76.3 ± 10.6 vs. 77.3 ± 10.8, $P < 0.0001$) were higher in the statin user group. Smoking history as never, ex-, or current was not statistically different between statin users and statin non-users. However, the statin non-user group showed a higher smoking amount than the statin user group (38.1% vs. 25.7% for less than a half pack, 14.3% vs. 23.3% for a pack to less than two packs, $P < 0.001$). In terms of drinking habits, the proportion of daily drinking (6.2% vs. 4.6%, $P = 0.004$) was higher in the statin non-user group, but the amount consumed at a time (50.1% vs. 43% for one to two drinks; 7.2% vs. 12.7% for five to six drinks; 5.7% vs. 7.3% for seven to nine drinks; $P = 0.001$) was lower. Interestingly, the frequency of no physical activity was higher (63.9% vs. 57.5%, $P < 0.0001$) and the mean frequency of vigorous (0.8 ± 1.7 vs. 0.9 ± 1.8, $P = 0.022$) and moderate (1.0 ± 1.9 vs. 1.2 ± 2.0, $P = 0.036$) physical activity per week were lower in the statin non-user group. The proportion of comorbidities of hypertension (14.7% vs. 18.7%, $P < 0.0001$) and heart disease (3.0% vs. 4.2%, $P = 0.026$) was higher in the statin user group, but that of cerebrovascular diseases (1.9% vs. 1.0%, $P = 0.013$) was higher in the statin non-user group.

The clinical outcomes of this study are shown in Table 2. Among 9,182 IPF patients analyzed during the study period, 850 were diagnosed as lung cancer. The incidence of lung cancer was similar in the statin user group and statin non-user group (9.2% vs. 9.4%, $P = 0.803$) during the study period. However, the duration from the date diagnosed with IPF to the development of lung cancer was significantly longer in the statin group (2,194.6 ± 1601.4 vs. 3,361.0 ± 1331.2, $P < 0.001$). Comparing the mortality rate between the statin user and non-user groups, OS was longer in the statin users (2413.9 ± 1778.7 vs. 3,741.8 ± 1443.1 days, Log rank $P < 0.0001$) and total number of deaths was significantly lower in (66.9% vs, 41.6%, hazard ratio [HR] 0.41, 95% CI 0.39–0.44, Log rank $P < 0.0001$). We obtained cumulative lung cancer incidence curves of stain users and statin non-users based on Kaplan-Meier plot analysis (Fig 1).

In multivariate analysis using a Cox regression model for lung cancer development in IPF patients, higher statin compliance (adjusted HR [aHR] 0.66, 95% CI 0.48–0.90, $P < 0.001$), statin use (aHR 0.63, 95% CI 0.53–0.76, $P < 0.001$), and female sex (aHR 0.43, 95% CI 0.33–0.56, $P < 0.001$) were independently associated with reduced lung cancer development in IPF patients. In contrast, the risk of cancer development increased in the group of patients diagnosed with IPF at an older age (aHR 1.05, 95% CI 1.04–1.06, $P < 0.001$) and with smoking (aHR 1.55, 95% CI 1.39–1.72, $P < 0.001$) (Table 3).

We also analyzed the risk factors for mortality in IPF patients after adjusting for demographic variables using multivariate Cox regression (Table 4). Statin use (aHR 0.43, 95% CI

**Table 1.  Demographic characteristics of study groups by statin use.**

| Characteristics | Statin use | | P-value |
|---|---|---|---|
| | No (n = 5810) | Yes (n = 3372) | |
| **Age at diagnosis of IPF** | 67.2 ± 11.2 | 64.0 ± 10.2 | < 0.001 |
| **Sex** | | | < 0.001 |
| Male | 3896 (67.1%) | 2002 (59.4%) | |
| Female | 1914 (32.9%) | 1370 (40.6%) | |
| **BMI** | 23.0 ± 3.2 | 24.0 ± 3.1 | < 0.001 |
| **Total cholesterol** | 184.2 ± 34.8 | 199.7 ± 42.6 | < 0.0001 |
| **Blood pressure** | | | |
| Systolic | 125.2 ± 17.1 | 126.6 ± 16.8 | < 0.0001 |
| Diastolic | 76.3 ± 10.6 | 77.3 ± 10.8 | < 0.0001 |
| **Smoking history** | | | 0.076 |
| Never | 3409 (61.1%) | 2016 (62.3%) | |
| Ex-smoker | 1100 (19.7%) | 574 (17.7%) | |
| Current smoker | 1073 (19.2%) | 644 (19.9%) | |
| **Smoking amount** | | | < 0.001 |
| Less than a half pack | 258 (38.1%) | 95 (25.7%) | |
| Half pack to less than one pack | 312 (46.1%) | 180 (48.8%) | |
| One pack to less than two packs | 97 (14.3%) | 86 (23.3%) | |
| More than two packs | 10 (1.5%) | 8 (2.2%) | |
| **Smoking duration** | | | 0.217 |
| Less than five years | 43 (4.1%) | 16 (2.9%) | |
| Five to nine years | 41 (3.9%) | 23 (4.2%) | |
| 10 to 19 years | 146 (13.9%) | 66 (12.1%) | |
| 20 to 29 years | 187 (17.8%) | 120 (22.1%) | |
| 30 years or more | 634 (60.3%) | 319 (58.6%) | |
| **Drinking frequency** | | | 0.004 |
| None | 2363 (70.0%) | 1285 (69.3%) | |
| Less than once per month | 311 (9.2%) | 155 (8.4%) | |
| Once per month | 347 (10.3%) | 214 (11.5%) | |
| Once per week | 147 (4.4%) | 113 (6.1%) | |
| Daily | 210 (6.2%) | 86 (4.6%) | |
| **Drinking amount at a time** | | | 0.001 |
| One to two drinks | 499 (50.1%) | 241 (43%) | |
| Three to four drinks | 369 (37.0%) | 208 (37.1%) | |
| Five to six drinks | 72 (7.2%) | 71 (12.7%) | |
| Seven to nine drinks | 57 (5.7%) | 41 (7.3%) | |
| **Physical activity frequency** | | | < 0.0001 |
| None | 2143 (63.9%) | 1060 (57.5%) | |
| One to two days per week | 582 (17.3%) | 391 (21.2%) | |
| Three to four days per week | 274 (8.2%) | 171 (9.3%) | |
| Five to six days per week | 72 (2.1%) | 41 (2.2%) | |
| Seven days per week | 284 (8.5%) | 179 (9.7%) | |
| **Physical activity intensity (per week)** | | | |
| Vigorous | 0.8 ± 1.7 | 0.9 ± 1.8 | 0.022 |
| Moderate | 1.0 ± 1.9 | 1.2 ± 2.0 | 0.036 |
| Walk | 2.5 ± 2.6 | 2.6 ± 2.6 | 0.156 |

(*Continued*)

**Table 1.** (Continued)

| Characteristics | Statin use | | P-value |
|---|---|---|---|
| | No (n = 5810) | Yes (n = 3372) | |
| **Comorbidities** | | | |
| **Liver disease** | 65 (1.9%) | 45 (2.4%) | 0.258 |
| **Hypertension** | 508 (14.7%) | 353 (18.7%) | < 0.0001 |
| **Cerebrovascular diseases** | 67 (1.9%) | 19 (1.0%) | 0.013 |
| **Heart disease** | 103 (3.0%) | 79 (4.2%) | 0.026 |
| **Diabetes** | 241 (7.0%) | 158 (8.4%) | 0.073 |
| **Other cancers** | 58 (1.7%) | 19 (1.0%) | 0.064 |

Data are presented as n (%) or mean ± SD.

IPF; Idiopathic pulmonary fibrosis, BMI; Body mass index.

0.39–0.46, $P<0.001$), female sex (aHR 0.67, 95% CI 0.61–0.73, $P<0.001$), higher physical activity frequency (aHR 0.93, 95% CI 0.90–0.96, $P<0.001$), and diabetes (aHR 0.78, 95% CI 0.69–0.88, $P<0.001$) were associated with reduced risk of mortality in IPF patients. In contrast, older age at IPF diagnosis (aHR 1.07, 95% CI 1.07–1.08, $P<0.001$) and smoking history (aHR 1.06, 95% CI 1.01–1.11, $P = 0.0182$) were associated significantly with shorter OS in IPF patients.

## Discussion

We identified clinical impacts of regular consecutive statin use in IPF patients who had both delayed lung cancer and prolonged OS. In Cox proportional hazard regression models, higher statin compliance, statin use, and female sex were independently associated with reduced risk of lung cancer, though older age at diagnosis of IPF and smoking history were associated with higher risk of lung cancer in IPF patients. For OS, statin use, female sex, higher physical activity frequency, and diabetes were associated with longer survival. In contrast, older age at diagnosis of IPF and smoking history were associated with shorter OS in IPF patients.

Aside from the well-known lipid-lowering effect, statins were reported to have anti-cancer effects through various pathways including inhibition of inflammation, immunomodulation, and angiogenesis [7]. Recently, the long-term use of statins was reported to reduce the risk of mortality in patients with lung cancer [34]. In a meta-analysis, statin use after diagnosis of lung cancer had a survival benefit for OS (HR 0.68, 95% CI 0.51–0.92) compared to those using statins before diagnosis [35]. Statins were also associated with prolonged survival in non-small cell lung cancer (NSCLC) patients treated with epidermal growth factor receptor

**Table 2. Comparison of clinical outcomes in IPF patients with and without statin use.**

| | Statin non-user (n = 5810) | Statin user (n = 3372) | P-value |
|---|---|---|---|
| **Lung cancer development in IPF patients (n = 850)** | 534 (9.2%) | 316 (9.4%) | 0.803 |
| **Duration (days)** | | | |
| **Lung cancer development** | 2194.6 ± 1601.4 | 3361.0 ± 1331.2 | < 0.0001 |
| **IPF diagnosis to death** | 2413.9 ±1778.7 | 3741.8 ± 1443.1 | < 0.0001 |
| **No. of deaths** | 3887 (66.9%) | 1404 (41.6%) | < 0.0001 |

Data are presented as n (%) or mean ± SD.

IPF = idiopathic pulmonary fibrosis.

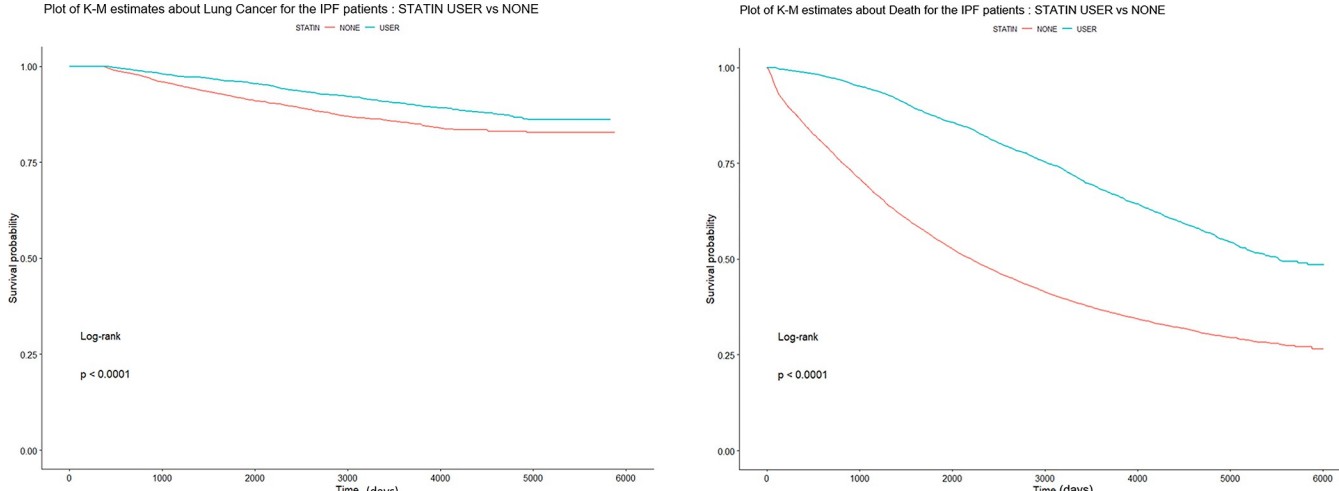

**Fig 1.** Kaplan-Meier plot displaying cumulative incidence of lung cancer development (A) and overall survival (B) in IPF patients with and without statin use. IPF = idiopathic pulmonary fibrosis, K-M = Kaplan-Meier.

tyrosine kinase inhibitor (EGFR-TKI)s [36]. Furthermore, statins can overcome EGFR-TKI resistance in patients with lung cancer harboring KRAS mutation, and they provided an increased response rate in lung cancer patients previously treated with nivolumab [37,38]. In contrast, there was no significant difference in efficacy between a group with addition of sim-vastatin to afatinib and a group with afatinib alone in patients with non-adenocarcinomatous NSCLC [39].

**Table 3. Cox proportional hazard regression analysis of the clinical variables affecting lung cancer development in IPF patients.**

| | aHR | Lower .95 | Upper .95 | *P*-value |
|---|---|---|---|---|
| **Statin use** | 0.6329 | 0.5260 | 0.7614 | < 0.001 |
| **Sex–female** | 0.4311 | 0.3332 | 0.5576 | < 0.001 |
| **Age at first diagnosis of IPF** | 1.0455 | 1.0357 | 1.0554 | < 0.001 |
| **BMI** | 1.0057 | 0.9762 | 1.0362 | 0.707 |
| **Total cholesterol** | 1.0009 | 0.9990 | 1.0028 | 0.366 |
| **Blood pressure** | | | | |
| Systolic | 0.9995 | 0.9922 | 1.0069 | 0.896 |
| Diastolic | 0.9995 | 0.9880 | 1.0111 | 0.934 |
| **Smoking history (ex + current)** | 1.5460 | 1.3911 | 1.7183 | < 0.001 |
| **Alcohol frequency** | 0.9982 | 0.9315 | 1.0696 | 0.959 |
| **physical activity frequency** | 0.9537 | 0.8886 | 1.0235 | 0.188 |
| **Comorbidities** | | | | |
| Liver diseases | 1.2332 | 0.6567 | 2.3158 | 0.514 |
| Hypertension | 1.0478 | 0.8229 | 1.3342 | 0.705 |
| Stroke | 1.7280 | 0.7670 | 3.8930 | 0.187 |
| Heart diseases | 1.0514 | 0.6454 | 1.7129 | 0.84 |
| Diabetes | 1.0218 | 0.7330 | 1.4246 | 0.899 |
| Cancers | 0.7224 | 0.3960 | 1.3180 | 0.289 |

aHR = adjusted hazard ratio, IPF = idiopathic pulmonary fibrosis, BMI = body mass index.

**Table 4. Cox proportional hazard regression analysis of the clinical variables affecting mortality in IPF patients.**

|  | aHR | Lower .95 | Upper .95 | *P*-value |
|---|---|---|---|---|
| **Statin use** | 0.4254 | 0.3919 | 0.4617 | < 0.001 |
| **Sex–female** | 0.6668 | 0.6121 | 0.7264 | < 0.001 |
| **Ages at first diagnosis of IPF** | 1.0714 | 1.0673 | 1.0756 | < 0.001 |
| **BMI** | 1.0020 | 0.9902 | 1.0139 | 0.7414 |
| **Total cholesterol** | 1.0005 | 0.9996 | 1.0014 | 0.2637 |
| **Blood pressure** |  |  |  |  |
| Systolic | 0.9996 | 0.9968 | 1.0024 | 0.7931 |
| Diastolic | 0.9999 | 0.9954 | 1.0044 | 0.959 |
| **Smoking history (ex + current)** | 1.0580 | 1.0096 | 1.1087 | 0.0182 |
| **Drinking experiences** | 0.9884 | 0.9586 | 1.0190 | 0.453 |
| **Physical activity frequency** | 0.9291 | 0.9028 | 0.9563 | < 0.001 |
| **Comorbidities** |  |  |  |  |
| Liver diseases | 0.9104 | 0.7143 | 1.1604 | 0.4482 |
| Hypertension | 1.0847 | 0.9868 | 1.1924 | 0.0921 |
| Stroke | 0.9767 | 0.7646 | 1.2477 | 0.8504 |
| Heart diseases | 0.9949 | 0.8323 | 1.1891 | 0.9548 |
| Diabetes | 0.7783 | 0.6885 | 0.8798 | < 0.001 |
| Cancers | 1.0853 | 0.8276 | 1.4233 | 0.5539 |

aHR = Adjusted hazard ratio, IPF = idiopathic pulmonary fibrosis, BMI = body mass index.

IPF has a progressive clinical course including a decline in pulmonary function, decrease in vital capacity, and diffusing capacity for carbon monoxide (DLCO) [40]. Further, lung cancer is a common morbidity of IPF with a prevalence of 4.4% to 13% [41]. If lung cancer develops in IPF patients, treatment modalities are limited regardless of lung cancer stage. The treatment goal for early-stage resectable lung cancer is complete remission. Standard curative treatment for patients with NSCLC is lobectomy [42]. Lung resection including lobectomy can cause a reduction in lung function, acute exacerbation (AE) of IPF, and acute respiratory distress syndrome (ARDS) [43]. For patients who are not surgical candidates due to medical reasons (e.g., cardiac or pulmonary failure), stereotactic ablative radiotherapy is a potential treatment option with comparable efficacy to surgery [44]. However, severe pulmonary toxicity such as radiation pneumonitis or ARDS was reported in 1.5–20% of patients who received stereotactic ablative radiotherapy in lung cancer patients with IPF [45]. Drug pneumonitis and IPF AE often recur in the advanced stages of lung cancer in IPF patients [46,47]. In a retrospective study, the incidence of lung cancer was reduced in IPF patients treated with pirfenidone [48]. However, no definite conclusion can be drawn from that retrospective study. For these reasons, strategies for early diagnosis and prevention of lung cancer are needed in IPF patients.

In our study, stain use was associated with delayed time from IPF diagnosis to lung cancer development. In a randomized controlled trial, there was a lower forced vital capacity (FVC) decline in IPF patients who received statins at baseline versus those who did not [49]. Also, statin use attenuated the decline in lung function in the elderly, and the effect of statins was estimated to be beneficial regardless of smoking status even though the size of the improvement varied among smoking groups [19]. In IPF patients, risk factors for lung cancer included being male, current smoking at IPF diagnosis, and rapid annual decline of 10% or more in FVC [40]. Decreased lung function is linked to increased inflammation and oxidative stress, and anti-inflammatory properties of statins were investigated in respiratory disease. In an

animal study, statins reduced neutrophil levels in lung tissue damaged by lipopolysaccharides [50]. Also, statins protected against smoking-induced lung damage and showed anti-inflammatory effects on the lung [51]. In lung transplant recipients, the levels of neutrophils and lymphocytes in the bronchoalveolar lavage of statin users were reduced compared with nonusers [52]. The inhibitory effect of statin on Ras farnesylation was well investigated. Kras alleles are activated in human lung adenocarcinomas, and inhibition of this is important in lung cancer prevention [53]. Also, lovastatin inhibits cell proliferation, cell cycle progression, and apoptosis in NSCLC cells through minichromosome maintenance (MCM) 2, involved in G1/S cell cycle inhibition [54]. Inflammation affects many aspects of malignancy including the proliferation and survival of cancer cells, angiogenesis, tumor metastasis, and tumor response to chemotherapeutic drugs [55]. The exact mechanism of the preventive effect of statin on lung cancer development in IPF patients is not fully understood, but it is believed that anti-inflammation actions on fibrotic lungs and the resulting lower decline in lung function may delay the occurrence of lung cancer. However, in meta-analysis, non-significant decrease of total lung cancer risk was observed among all statin users (RR = 0.89, 95% CI 0.78–1.02) [56]. Further randomized controlled trials and high quality cohort studies are needed to confirm this association.

In our study, statin users had lower risk of death among IPF patients. This finding is consistent with a previous study. Kreuter et al. reported that statins might have a beneficial effect on the clinical outcomes of IPF patients including lower risks of death, six-minute walk distance decline, all-cause hospitalization, and IPF-related mortality [18]. Repetitive alveolar epithelial injury triggered the early development of IPF. The exact etiology of IPF is unknown, and all stages of fibrosis are accompanied by innate and adaptive immune responses [57]. Modulatory effects of statins on pathways of fibrosis were investigated in vitro studies. Exposure to statins resulted in a reversible and time-dependent change in cell morphology in human renal fibroblasts [58]. Fluvastatin inhibits TGF-ß1-induced thrombospondin-1 expression in coronary artery smooth muscle cells [59]. However, statin use and all-cause mortality in IPF patients showed controversial results based on a statistical analysis [60]. A prospective cohort study with dosage of statin, statin adherence, and use of concurrent antifibrotics is needed to confirm beneficial effects of statin therapy in IPF patients.

In our study, higher physical activity frequency decreased all-cause mortality by 8% in IPF patients. It is well known that cardiopulmonary exercise tests and six-min walk tests provide prognostic value of mortality in patients with IPF [61]. Decreased physical activity was associated with lower progression-free survival (HR 12.1, 95% CI, 1.9–78.8, $P = 0.009$) in IPF patients. Lower quadriceps strength and higher depression scores contribute to lower physical activity [62]. Pulmonary rehabilitation using exercise training is effective for improving exercise capacity, dyspnea, and quality of life in IPF patients [63]. Also, pulmonary rehabilitation noncompletion and nonresponse were associated independently with increased one-year all-cause mortality in IPF patients [64]. Even without active intervention such as respiratory rehabilitation, life style behaviors such as shorter daily sitting and longer weekly walking were associated with reduced hospitalization and mortality risks in patients with IPF [65]. Although the physical activity frequency of our study was collected using a subjective self-report, it supports the previous study findings that exercise can lower the mortality of IPF patients based on large-scale cohort data.

Interestingly, not only did the physical activity frequency reduce all-cause mortality, but comorbid diabetes showed a 22% risk reduction of death in IPF patients. The biology of aging may influence the susceptibility to lung fibrosis in the elderly, increasing the incidence of IPF in patients over 60 years of age [66]. Relatively older populations with IPF have variable comorbidities such as hypertension, cardiovascular diseases, and diabetes. Type 2 diabetes mellitus (DM) is a common underlying disease in many IPF patients [67]. DM is a systemic

metabolic disease characterized by persistent hyperglycemia, and the lungs are targeted by diabetic micro-vascular damage [68]. Epidemiological research reported that diabetes is a risk factor for IPF, with the prevalence of IPF accompanied by DM estimated to be 10–42% even when excluding cases treated with glucocorticoids [69,70]. Further, DM was reported to be a risk factor with higher mortality in an IPF population (HR 2.5, 95% CI 1.04–5.9) [71]. Contrary to the prior study, our study suggested that DM was associated with reduced risk of mortality in IPF patients. This may be partly due to diabetic medications. Metformin is the first-choice of treatment for glycemic control [72]. Aside from the glucose lowering effect, metformin was involved in anti-fibrotic physiology associated with AMPK activation and showed an inhibitory effect in myofibroblasts differentiation [73]. Also, GLP-1 receptor agonists were found to have anti-pulmonary fibrotic effects and alleviated bleomycin-induced lung damage and fibrosis through inactivation of nuclear factor kappa-B in animal studies [74]. In our study, we did not conduct an investigation of diabetic drugs, so it was not possible to confirm whether mortality was reduced by DM or diabetic drugs. Further investigations through a survey on individualized diabetic medication intake are necessary to determine the effect of DM on mortality in IPF patients.

The limitations of our study should be recognized. First, it had been lead to the possibility of the reduced or increased numbers populations because of the assignment of IPF was by multiple healthcare providers using the ICD-10 code. Second, as the study design was retrospective and based on a large population-based cohort, there was the possibility of selection bias of confounding factors that might have influenced the study results. Third, we tried to include drug compliance, but the true medication adherence could not be estimated. Instead, we applied a mathematical equation of drug compliance based on the total days of statin prescribed divided by the study period. Forth, the dose and the different potency of statin were not included as confounding factors. Fifth, we did not consider antifibrotics (including pirfenidone and nintedanib), which affect disease progression and mortality in IPF patients, as confounding factors. This is because the timing of the introduction of the drug into country and the timing of the reimbursement did not coincide with the time of data collection. Lastly, we did not include the severity and status of IPF based on pulmonary function (e.g., FVC, DLCO), 6 minute walk test and proportions of fibrotic tissues (honeycombing) on imaging, because NHIS dataset does not collect these variables. To identify further effects of statin use on lung cancer development and mortality in IPF patients, a well-designed large scale prospective study is necessary.

## Conclusions

The goal of this study was to assess the clinical impact of consecutive statin administration on the development of lung cancer and OS in patients with IPF using the NHIS database. The findings of this study showed that consecutive statin use delayed the onset of lung cancer in IPF patients and improved OS rates. Considering the higher prevalence of comorbidities such as diabetes and hypertension, in IPF patients, addition of statin would be beneficial for clinical outcomes. A well-designed large-scale cohort study is needed to confirm the beneficial effects of statin use on preventing lung cancer development and reduced risk of mortality in IPF patients.

## Supporting information

**S1 Appendix. Codes list of interstitial lung disease.**
(DOCX)

## Acknowledgments

Informed consent was waived because data analysis was performed retrospectively using anonymized data derived from the NHIS in Korea.

## Author Contributions

**Conceptualization:** Yoo Jung Lee, Hye Seon Kang.

**Data curation:** Yoo Jung Lee, Nayoon Kang, Jiwon Ryoo, Hye Seon Kang.

**Formal analysis:** Nayoon Kang, Hye Seon Kang.

**Funding acquisition:** Hye Seon Kang.

**Investigation:** Yoo Jung Lee, Nayoon Kang, Yong Hyun Kim, Hye Seon Kang.

**Methodology:** Nayoon Kang, Hye Seon Kang.

**Project administration:** Yoo Jung Lee, Yong Hyun Kim, Hye Seon Kang.

**Resources:** Nayoon Kang, Hye Seon Kang.

**Software:** Nayoon Kang.

**Supervision:** Soon Seog Kwon, Yong Hyun Kim, Hye Seon Kang.

**Validation:** Nayoon Kang, Junghyun Nam, Yong Hyun Kim, Hye Seon Kang.

**Visualization:** Nayoon Kang.

**Writing – original draft:** Yoo Jung Lee, Eung Gu Lee, Jiwon Ryoo, Yong Hyun Kim, Hye Seon Kang.

**Writing – review & editing:** Yoo Jung Lee, Soon Seog Kwon, Yong Hyun Kim, Hye Seon Kang.

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
