## [Decision Letter · Decision Letter 0]

2 Nov 2023

PONE-D-23-28368The preventative effects of statin on lung cancer development in patients with idiopathic pulmonary fibrosis using the National Health Insurance Service Database in KoreaPLOS ONE

Dear Dr. Kang,

Thank you for submitting your manuscript to PLOS ONE. After careful consideration, we feel that it has merit but does not fully meet PLOS ONE’s publication criteria as it currently stands. Therefore, we invite you to submit a revised version of the manuscript that addresses the points raised during the review process.

We look forward to receiving your revised manuscript.

Kind regards,

Tsai-Ching Hsu, Ph.D.

Academic Editor

PLOS ONE

Journal Requirements:

This work was supported by the Institute of Clinical Medicine Research of Bucheon St. Mary’s Hospital, Research Fund, 2020 (BCMC20BD07). 

Additional Editor Comments:

Please revise your manuscript according to the referees’ comments and ensure the revised manuscript with all changes clearly highlighted.

Reviewers' comments:

Reviewer's Responses to Questions

**Comments to the Author**

1. Is the manuscript technically sound, and do the data support the conclusions?

Reviewer #1: Yes

Reviewer #2: Yes

2. Has the statistical analysis been performed appropriately and rigorously? 

Reviewer #1: No

Reviewer #2: No

3. Have the authors made all data underlying the findings in their manuscript fully available?

Reviewer #1: No

Reviewer #2: Yes

4. Is the manuscript presented in an intelligible fashion and written in standard English?

Reviewer #1: Yes

Reviewer #2: Yes

5. Review Comments to the Author

Reviewer #1: The authors present an epidemiological study with a satisfying sample of patients regarding the impact of statin use in IPF survival. However, there are methodological issues that may have affected research results. Particularly, as the authors do not include lung function parameters and the use of antifibrotics in their registry, important aspects that is known to affect survival in IPF studies limit the significance of their results.

Major comments:

1. You do not include important clinical parameters in IPF patients of both groups, including FVC, DLCO and 6MWT that could have impacted the survival. Lung function would allow us to assess the severity of both groups. I think that this is a serious limitation in the study.

2. The fact that the exercise was significantly different between the statin and the non statin groups could be related to the severity of patients. In case of severe pulmonary fibrosis in which patients need oxygen may have less exercise that other milder cases.

3. The are diverse clinical characteristics that differ between the 2 groups.

Why you do not include all these variables in the multivariate analysis.

4. Did you include age in the multivariate analysis?

5. How could diabetes have affected OS in IPF? Maybe there is a confounding here?

6. Another important limitation is the fact that you have not taken into consideration antifibrotics that affect progression and mortality. So, this may have also affect OS threatening your internal validity.

7. Another limitation, is that you have included patients based on ICD-10, here there might be also misclassification and information bias. It should be also stated in the limitations paragraph.

Minor comments:

1. Please correct in Table 2, 1st row, 3rd column the word ‘statin’.

Reviewer #2: This study shows the clinical impact of statin administration on the development of lung cancer and overall survival in patients with IPF using the NHIS database. However, the following comments are need to be modified.

Major>

1. Please provide the "Patients flow chart" as figure 1 including inclusion criteria and exclusion criteria.

2. Did you conduct multivariate analysis in Table 3? The variable 'statin compliance' applies only to the 'statin user' group, so does that mean the non-user group was excluded from the analysis?

3. You previously suggested that statins have a preventive effect on lung cancer development in IPF patients. However, this effect was not found in the current paper. What do you think about this? (https://doi.org/10.1016/j.chest.2022.08.1445)

Minor>

1. Please provide the disease name for the injury code as supplementary material.

(M05.1, M05.2,78 M05.3, M05.8, M05.9, M06.0, M06.8, M06.9, M30.1, M31.3, M31.7, M32, M33, M34, M35.0,

79 M35.1, D86, J84.0, J60~J70.9. J84 etc.)

2. Please provide a more detailed definition of “statin use”.

3. Please provide definitions of “Drinking amount at a time” and “Physical activity intensity” in the text.

6. PLOS authors have the option to publish the peer review history of their article (what does this mean?). If published, this will include your full peer review and any attached files.

Reviewer #1: No

Reviewer #2: No

---

## [Author Response · Author response to Decision Letter 0]

29 Dec 2023

Dear Editor-in-Chief, 

We would like to thank you and the reviewers of the PLOS ONE for taking the time to review our article. It is truly honorable to receive the letter of revision to enrich the work of research we have performed. We would like to appreciate for the time and efforts by the editors to this paper. We have made some corrections and clarifications in the manuscript after going over the reviewers’ comments. 

The changes are summarized below:

Reviewer #1:

The authors present an epidemiological study with a satisfying sample of patients regarding the impact of statin use in IPF survival. However, there are methodological issues that may have affected research results. Particularly, as the authors do not include lung function parameters and the use of antifibrotics in their registry, important aspects that is known to affect survival in IPF studies limit the significance of their results.

Major comments:

1. You do not include important clinical parameters in IPF patients of both groups, including FVC, DLCO and 6MWT that could have impacted the survival. Lung function would allow us to assess the severity of both groups. I think that this is a serious limitation in the study.

Answer: We agree with reviewer’s opinion. As the reviewer told, functional volume capacity (FVC), diffusing capacity for carbon monoxide (DLCO) and six-minute walk test (6MWT) are very important factors to impact on the survival of IPF patients. However, NHIS dataset does not include serial pulmonary function tests and 6MWT of every patient, therefore it was hardly possible to be gathering and analyze each of FVC and forced expiratory volume in one second (FEV1) of selected 9182 patients. We further described detailed sentences in discussion section, line 342-345.

2. The fact that the exercise was significantly different between the statin and the non-statin groups could be related to the severity of patients. In case of severe pulmonary fibrosis in which patients need oxygen may have less exercise than other milder cases.

Answer: Thank you for your comments. We also agree with reviewer’s comments. The the phrase of ‘physical activity’ and the ‘exercise’ as a noun were used interchangeably in Korean languege, so we made correction from the word of ‘exercise’ in the manuscript into ‘physical activity’ at table 1, table 3 and table 4, and highlighted the corrections in manuscript at line 35, 208-209, 224-225, 294, 305 and 308 because it was more appropriate. Physical activity frequency were different between statin and non-statin user group, and lower frequency of physical activity was noted at statin user group. As the reviewer pointed, we could not consider the severity of IPF based on pulmonary function test, 6MWT and oxygen use, because NHIS dataset does not collect these varibles, and we added detailed sentences in limitation, line 342-345. However, considering the possibility of correlation between the physical activity frequency and other clinical factors associated with IPF patients, we performed cox proportional hazard regression analysis including physical activity frequency as confounding factors. As a result, physical activity frequency was independently associated with reduced hazard in mortality of IPF patients, but not associated with lung cancer development. 

3. There are diverse clinical characteristics that differ between the 2 groups. Why you do not include all these variables in the multivariate analysis.

Answer: Thank you for your comments. We additionally included all variables for multivariate analysis (age at diagnosis of IPF, sex, BMI, total cholesterol, blood pressure, smoking history, drinking frequency, drinking amount, physical activity frequency, comorbidities) which are different between two groups (Table 3).

4. Did you include age in the multivariate analysis?

Answer: Thank you for your comments. We included ages at first diagnosis of IPF in multivariate analysis (Table 3 and Table 4).

5. How could diabetes have affected OS in IPF? Maybe there is a confounding here?

Answer: We agree with your reviewer’s opinion. In our study, diabetes mellitus (DM) was associated with reduced hazard ratio of OS in IPF patients. As we explained at discussion section, these findings would be associated to DM medications. Metformin is known to be involved in anti-fibrotic physiology and has inhibitory effect in myofibroblasts differentiations. Also, another popularly used GLP-1 receptor agonist was found to have anti-pulmonary fibrosis effects in animal models. In our study, we did not conduct an investigation on various types of diabetic drugs, so it was not possible to confirm whether mortality was reduced by DM or diabetic drugs. So, further investigation was needed to confirm these findings.

6. Another important limitation is the fact that you have not taken into consideration antifibrotics that affect progression and mortality. So, this may have also affected OS threatening your internal validity.

Answer: We fully agree with reviewer’s comments. We collected IPF patients between 2002 and 2018. The official date of release of pirfenidone was July 2011(https://www.shionogi.com/content/dam/shionogi/global/news/pdf/2011/e_110712.pdf). However, pirfenidone has been reimbursed since 2015, and nintedanib is still not reimbursed in Korea. The definite limitation of our study is that the history of two antifibrotic drugs were not included in the analysis because the timing of domestic use of drugs and imbursements were different. We added detailed sentences in line 341-342.

7. Another limitation, is that you have included patients based on ICD-10, here there might be also misclassification and information bias. It should be also stated in the limitations paragraph.

Answer: Thank you for your comments. We added detailed sentences at limitation section, line 331-333.

Minor comments:

1. Please correct in Table 2, 1st row, 3rd column the word ‘statin’.

Answer: Thank you for your comments. We corrected the word at table 2.

 

Reviewer #2: 

This study shows the clinical impact of statin administration on the development of lung cancer and overall survival in patients with IPF using the NHIS database. However, the following comments are need to be modified.

Major>

1. Please provide the "Patients flow chart" as figure 1 including inclusion criteria and exclusion criteria. 

Response: We fully agree with reviewer’s comments. We summarized briefly inclusion and exclusion criteria at material and method sections. There was an issue on processing the number of patients we selected and excluded in the flowchart because the period of use of health insurance data has expired, it is currently difficult to accurately determine the number of patients who meet the exclusion criteria. Instead, in order to clarify to understand out study, we added to adjunctive information in the manuscript about the study design and the population in the section of method. (line 76-86) 

2. Did you conduct multivariate analysis in Table 3? The variable 'statin compliance' applies only to the 'statin user' group, so does that mean the non-user group was excluded from the analysis?

Answer: The results shown on Table 3 by Cox proportional hazard analysis were performed under multivariate analysis. We agree with reviewer’s opinion, so variable ‘statin compliance’ was excluded from our multivariate analysis. (Table 3)

3. You previously suggested that statins have a preventive effect on lung cancer development in IPF patients. However, this effect was not found in the current paper. What do you think about this?

(https://doi.org/10.1016/j.chest.2022.08.1445)

Answer: Thank you for your review our other study. Previous study was conducted using our clinical ware house on the same topic as this study, and showed the same conclusions as current study using NHIS data. In previous study, taking statin reduced hazard ratio for lung cancer development and was associated with longer overall survival in IPF patients.

One of the results that we would like to emphasize on this study was the higher compliance with statin use had better risk reducing effect on lung cancer development by delaying its onset of time during the lifetime of the IPF patients. To confirm the preventative effect of regular statin use in lung cancer development, prospective study design should be followed to overcome the data-based results we performed. 

Minor>

1. Please provide the disease name for the injury code as supplementary material. (M05.1, M05.2,78 M05.3, M05.8, M05.9, M06.0, M06.8, M06.9, M30.1, M31.3, M31.7, M32, M33, M34, M35.0, 79 M35.1, D86, J84.0, J60~J70.9. J84 etc.)

Answer: Thank you so much for your comments. We made the list of disease for each codes and attached under the file named ‘S1_Codes list of interstitial lung diseases.docx’ and described on line 649.

2. Please provide a more detailed definition of “statin use”.

Answer: We’re grateful for your comments. The detailed description of “statin use” are added on the section of method in line 100-106 and revised its limitation in line 335-338.

3. Please provide definitions of “Drinking amount at a time” and “Physical activity intensity” in the text.

Answer: Thank you for the comments. The definitions of “Drinking amount at a time” was described in line 111-123. The definitions of “Physical activity intensity” were exhibited at line 125-134.

---

## [Decision Letter · Decision Letter 1]

1 Feb 2024

PONE-D-23-28368R1The preventative effects of statin on lung cancer development in patients with idiopathic pulmonary fibrosis using the National Health Insurance Service Database in KoreaPLOS ONE

Dear Dr. Kang,

Thank you for submitting your manuscript to PLOS ONE. After careful consideration, we feel that it has merit but does not fully meet PLOS ONE’s publication criteria as it currently stands. Therefore, we invite you to submit a revised version of the manuscript that addresses the points raised during the review process.

We look forward to receiving your revised manuscript.

Kind regards,

Tsai-Ching Hsu, Ph.D.

Academic Editor

PLOS ONE

Journal Requirements:

Reviewers' comments:

Reviewer's Responses to Questions

**Comments to the Author**

1. If the authors have adequately addressed your comments raised in a previous round of review and you feel that this manuscript is now acceptable for publication, you may indicate that here to bypass the “Comments to the Author” section, enter your conflict of interest statement in the “Confidential to Editor” section, and submit your "Accept" recommendation.

Reviewer #2: All comments have been addressed

Reviewer #3: (No Response)

2. Is the manuscript technically sound, and do the data support the conclusions?

Reviewer #2: Yes

Reviewer #3: Yes

3. Has the statistical analysis been performed appropriately and rigorously? 

Reviewer #2: Yes

Reviewer #3: Yes

4. Have the authors made all data underlying the findings in their manuscript fully available?

Reviewer #2: Yes

Reviewer #3: Yes

5. Is the manuscript presented in an intelligible fashion and written in standard English?

Reviewer #2: Yes

Reviewer #3: Yes

6. Review Comments to the Author

Reviewer #2: (No Response)

Reviewer #3: erratum:

1.line 223:

and older age at diagnosis

advice shifting " and " to "though " and it will make readers understand more easily

2.line 287

were invetigated by

Would you mind to get rid of the word "by " here?

3.Figure 1-A/1-B:

Would you mind to write ipf into IPF in capital letters?

4.line -41

Would you mind to change the words"lipid-lowering " into "cholesterol production"

it would be more correct , because that when statin blocks active site of lipid-lowering ezyme, how it will make lipid getting down possible by blocking its function

5.line 166:

maybe you miss the symbol of "% " behind the number 18.7

6.suggestion on your discussion:

Is it still the same results as well as the DM , a reduced mortality(OS) factor, in non-statin user group after your meta-analysis?

You may consider to support and strengthen your point of view if it still has significant P value in non-statin use group processing meta-analysis

7. PLOS authors have the option to publish the peer review history of their article (what does this mean?). If published, this will include your full peer review and any attached files.

Reviewer #2: No

Reviewer #3: No

---

## [Author Response · Author response to Decision Letter 1]

8 Feb 2024

Dear Editor-in-Chief, 

We would like to thank you and the reviewers of the PLOS ONE for taking the time to review our article. It is truly honorable to receive the letter of revision to enrich the work of research we have performed. We would like to appreciate for the time and efforts by the editors to this paper. We have made some corrections, answers, and clarifications in the manuscript after going over the reviewers’ comments. 

The changes are summarized below:

Reviewer #1:

The authors present an epidemiological study with a satisfying sample of patients regarding the impact of statin use in IPF survival. However, there are methodological issues that may have affected research results. Particularly, as the authors do not include lung function parameters and the use of antifibrotics in their registry, important aspects that is known to affect survival in IPF studies limit the significance of their results.

Major comments:

1. You do not include important clinical parameters in IPF patients of both groups, including FVC, DLCO and 6MWT that could have impacted the survival. Lung function would allow us to assess the severity of both groups. I think that this is a serious limitation in the study.

Answer: We agree with reviewer’s opinion. As the reviewer told, functional volume capacity (FVC), diffusing capacity for carbon monoxide (DLCO) and six-minute walk test (6MWT) are very important factors to impact on the survival of IPF patients. However, NHIS dataset does not include serial pulmonary function tests and 6MWT of every patient, therefore it was hardly possible to be gathering and analyze each of FVC and forced expiratory volume in one second (FEV1) of selected 9182 patients. We further described detailed sentences in discussion section, line 342-345.

2. The fact that the exercise was significantly different between the statin and the non-statin groups could be related to the severity of patients. In case of severe pulmonary fibrosis in which patients need oxygen may have less exercise than other milder cases.

Answer: Thank you for your comments. We also agree with reviewer’s comments. The the phrase of ‘physical activity’ and the ‘exercise’ as a noun were used interchangeably in Korean languege, so we made correction from the word of ‘exercise’ in the manuscript into ‘physical activity’ at table 1, table 3 and table 4, and highlighted the corrections in manuscript at line 35, 208-209, 224-225, 294, 305 and 308 because it was more appropriate. Physical activity frequency were different between statin and non-statin user group, and lower frequency of physical activity was noted at statin user group. As the reviewer pointed, we could not consider the severity of IPF based on pulmonary function test, 6MWT and oxygen use, because NHIS dataset does not collect these varibles, and we added detailed sentences in limitation, line 342-345. However, considering the possibility of correlation between the physical activity frequency and other clinical factors associated with IPF patients, we performed cox proportional hazard regression analysis including physical activity frequency as confounding factors. As a result, physical activity frequency was independently associated with reduced hazard in mortality of IPF patients, but not associated with lung cancer development. 

3. There are diverse clinical characteristics that differ between the 2 groups. Why you do not include all these variables in the multivariate analysis.

Answer: Thank you for your comments. We additionally included all variables for multivariate analysis (age at diagnosis of IPF, sex, BMI, total cholesterol, blood pressure, smoking history, drinking frequency, drinking amount, physical activity frequency, comorbidities) which are different between two groups (Table 3).

4. Did you include age in the multivariate analysis?

Answer: Thank you for your comments. We included ages at first diagnosis of IPF in multivariate analysis (Table 3 and Table 4).

5. How could diabetes have affected OS in IPF? Maybe there is a confounding here?

Answer: We agree with reviewer’s opinion. In our study, diabetes mellitus (DM) was associated with reduced hazard ratio of OS in IPF patients. As we explained at discussion section, these findings would be associated to DM medications. Metformin is known to be involved in anti-fibrotic physiology and has inhibitory effect in myofibroblasts differentiations. Also, another popularly used GLP-1 receptor agonist was found to have anti-pulmonary fibrosis effects in animal models. In our study, we did not conduct an investigation on various types of diabetic drugs, so it was not possible to confirm whether mortality was reduced by DM or diabetic drugs. So, further investigation was needed to confirm these findings.

6. Another important limitation is the fact that you have not taken into consideration antifibrotics that affect progression and mortality. So, this may have also affected OS threatening your internal validity.

Answer: We fully agree with reviewer’s comments. We collected IPF patients between 2002 and 2018. The official date of release of pirfenidone was July 2011(https://www.shionogi.com/content/dam/shionogi/global/news/pdf/2011/e_110712.pdf). However, pirfenidone has been reimbursed since 2015, and nintedanib is still not reimbursed in Korea. The definite limitation of our study is that the history of two antifibrotic drugs were not included in the analysis because the timing of domestic use of drugs and imbursements were different. We added detailed sentences in line 341-342.

7. Another limitation, is that you have included patients based on ICD-10, here there might be also misclassification and information bias. It should be also stated in the limitations paragraph.

Answer: Thank you for your comments. We added detailed sentences at limitation section, line 331-333.

Minor comments:

1. Please correct in Table 2, 1st row, 3rd column the word ‘statin’.

Answer: Thank you for your comments. We corrected the word at table 2.

 

Reviewer #2: 

This study shows the clinical impact of statin administration on the development of lung cancer and overall survival in patients with IPF using the NHIS database. However, the following comments are need to be modified.

Major>

1. Please provide the "Patients flow chart" as figure 1 including inclusion criteria and exclusion criteria. 

Response: We fully agree with reviewer’s comments. We summarized briefly inclusion and exclusion criteria at material and method sections. There was an issue on processing the number of patients we selected and excluded in the flowchart because the period of use of health insurance data has expired, it is currently difficult to accurately determine the number of patients who meet the exclusion criteria. Instead, in order to clarify to understand out study, we added to adjunctive information in the manuscript about the study design and the population in the section of method. (line 76-86) 

2. Did you conduct multivariate analysis in Table 3? The variable 'statin compliance' applies only to the 'statin user' group, so does that mean the non-user group was excluded from the analysis?

Answer: The results shown on Table 3 by Cox proportional hazard analysis were performed under multivariate analysis. We agree with reviewer’s opinion, so variable ‘statin compliance’ was excluded from our multivariate analysis. (Table 3)

3. You previously suggested that statins have a preventive effect on lung cancer development in IPF patients. However, this effect was not found in the current paper. What do you think about this?

(https://doi.org/10.1016/j.chest.2022.08.1445)

Answer: Thank you for your review to our other study. Previous study was conducted using our clinical warehouse on the same topic as this study, and showed the same conclusions as current study using NHIS data. In previous study, taking statin reduced hazard ratio for lung cancer development and was associated with longer overall survival in IPF patients.

One of the results that we would like to emphasize on this study was the higher compliance with statin use had better risk reducing effect on lung cancer development by delaying its onset of time during the lifetime of the IPF patients. To confirm the preventative effect of regular statin use in lung cancer development, prospective study design should be followed to overcome the data-based results we performed. 

Minor>

1. Please provide the disease name for the injury code as supplementary material. (M05.1, M05.2,78 M05.3, M05.8, M05.9, M06.0, M06.8, M06.9, M30.1, M31.3, M31.7, M32, M33, M34, M35.0, 79 M35.1, D86, J84.0, J60~J70.9. J84 etc.)

Answer: Thank you so much for your comments. We made the list of disease for each codes and attached under the file named ‘S1_Codes list of interstitial lung diseases.docx’ and described on line 649.

2. Please provide a more detailed definition of “statin use”.

Answer: We’re grateful for your comments. The detailed description of “statin use” are added on the section of method in line 100-106 and revised the limitations of our study in the third section, specifically in lines 335-338.

3. Please provide definitions of “Drinking amount at a time” and “Physical activity intensity” in the text.

Answer: Thank you for the comments. The definitions of “Drinking amount at a time” was described in line 111-123. The definitions of “Physical activity intensity” were exhibited at line 125-134. 

 

Reviewer #3:

1. ~ 5. Erratum:

Answer: The errata suggested from No.1 to 5 were mostly made corrections on the manuscript and highlighted; No.1 was in line 223, No.2 was in line 287, No.4 was in line 41, and No.5 was in line 166. Lastly, in response to the erratum No.3, the lowercase letters “ipf” have been corrected to capital letters in the description of Figure 1-A and 1-B and uploaded. Thank you for your comments.

6. Is it still the same results as well as the DM, a reduced mortality (OS) factor in non-statin user group after your meta-analysis? You may consider to support and strengthen your point of view if it still has significant P value in non-statin use group processing meta-analysis.

Answer: Thank you for the suggestion. In our study, DM was identified as one of the risk-reducing factors for overall survival (OS) in IPF patients, encompassing both statin users and non-users, with a reduction of nearly 22%. According to the multivariate Cox regression analysis, we proposed that the influencing factor may be attributed to glucose-lowering drugs, such as metformin, exhibiting potent anti-fibrotic effects. Further investigation is necessary to ascertain the prescription rates, types, and effects of diabetes medications on IPF mortality in both statin and non-statin users in the future.

---

## [Decision Letter · Decision Letter 2]

12 Feb 2024

The preventative effects of statin on lung cancer development in patients with idiopathic pulmonary fibrosis using the National Health Insurance Service Database in Korea

PONE-D-23-28368R2

Dear Dr. Kang,

We’re pleased to inform you that your manuscript has been judged scientifically suitable for publication and will be formally accepted for publication once it meets all outstanding technical requirements.

Kind regards,

Tsai-Ching Hsu, Ph.D.

Academic Editor

PLOS ONE

Additional Editor Comments (optional):

Reviewers' comments:

Reviewer's Responses to Questions

**Comments to the Author**

1. If the authors have adequately addressed your comments raised in a previous round of review and you feel that this manuscript is now acceptable for publication, you may indicate that here to bypass the “Comments to the Author” section, enter your conflict of interest statement in the “Confidential to Editor” section, and submit your "Accept" recommendation.

Reviewer #3: (No Response)

2. Is the manuscript technically sound, and do the data support the conclusions?

Reviewer #3: Yes

3. Has the statistical analysis been performed appropriately and rigorously? 

Reviewer #3: Yes

4. Have the authors made all data underlying the findings in their manuscript fully available?

Reviewer #3: Yes

5. Is the manuscript presented in an intelligible fashion and written in standard English?

Reviewer #3: No

6. Review Comments to the Author

Reviewer #3: (No Response)

7. PLOS authors have the option to publish the peer review history of their article (what does this mean?). If published, this will include your full peer review and any attached files.

Reviewer #3: No

---

## [Editor Report · Acceptance letter]

26 Feb 2024

PONE-D-23-28368R2 

PLOS ONE

Dear Dr. Kang, 

I'm pleased to inform you that your manuscript has been deemed suitable for publication in PLOS ONE. Congratulations! Your manuscript is now being handed over to our production team.

Kind regards, 

on behalf of

Dr. Tsai-Ching Hsu 

Academic Editor

PLOS ONE